# Revisiting Source Context in Nearest Neighbor Machine Translation

**Xuanhong Li**[1,2,3] , **Peng Li**[4,‡] , **Po Hu**[1,2,3,‡]

[1] Hubei Provincial Key Laboratory of Artificial Intelligence and Smart Learning,
Central China Normal University, Wuhan, Hubei, China
[2] School of Computer Science, Central China Normal University, Wuhan, Hubei, China
[3] National Language Resources Monitoring & Research Center for Network Media,
Central China Normal University, Wuhan, Hubei, China
[4] Institute for AI Industry Research (AIR), Tsinghua University, Beijing, China
xuanhong.li@mails.ccnu.edu.cn, lipeng@air.tsinghua.edu.cn
phu@mail.ccnu.edu.cn

## Abstract

Nearest neighbor machine translation ($k$NN-MT), which interpolates target token probabilities with estimates derived from additional examples, has achieved significant improvements and attracted extensive interest in recent years. However, existing research does not explicitly consider the source context when retrieving similar examples, potentially leading to suboptimal performance. To address this, we comprehensively revisit the role of source context and propose a simple and effective method for improving neural machine translation via source context enhancement, demonstrating its crucial role in both retrieving superior examples and determining more suitable interpolation coefficients. Furthermore, we reveal that the probability estimation can be further optimized by incorporating a source-aware distance calibration module. Comprehensive experiments show that our proposed approach can be seamlessly integrated with representative $k$NN-MT baselines, resulting in substantial improvements over these strong baselines across a number of settings and domains. Remarkably, these improvements can reach up to 1.6 BLEU points.[1]

## 1 Introduction

Nearest neighbor machine translation (Khandelwal et al., 2021, $k$NN-MT) enhances conventional neural machine translation (NMT) by adopting a retrieval-based strategy to interpolate the target token probability distribution produced by the NMT model with the one estimated from similar examples of an auxiliary datastore. Since demonstrating promising results in a wide range of machine translation scenarios such as multilingual translation (Khandelwal et al., 2021; Li et al., 2022), do-

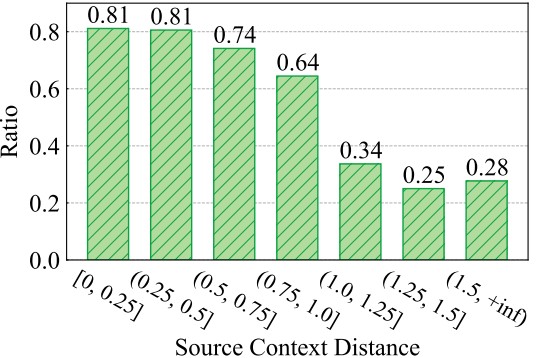

Figure 1: The ratio of the top-$k$ retrieved examples whose value is equal to the ground-truth token, which is roughly inversely correlated with the source context distance, indicating the potential benifit of source context for $k$NN-MT. The development set of IT domain from (Zheng et al., 2021a) is used and $k = 32$. Please refer to introduction for more detials.

main adaptation (Zheng et al., 2021b; Du et al., 2022), and online learning (Wang et al., 2022b), it has attracted increasing attention in recent years.

Due to its success, extensive efforts have been made to improve the effectiveness of $k$NN-MT. Zheng et al. (2021a) propose a dynamic approach to determine the number of examples to be retrieved. In contrast, Wang et al. (2022c) concentrate on developing retrieval-aware example representations. Jiang et al. (2022) suggest leveraging NMT predicted results to calibrate $k$NN distribution. Additionally, Zheng et al. (2021a) improve the estimation of probability distribution by employing an ensemble of multiple estimations derived from different numbers of examples. Despite these advances, the source context, e.g. the source side of a parallel sentence pair, has not been explicitly considered in these works.

Preliminary study shows that source context may have significant potiential in $k$NN-MT. Typically,

‡ Peng Li (lipeng@air.tsinghua.edu.cn) and Po Hu (phu@mail.ccnu.edu.cn) are corresponding authors.

[1]Our code is available at https://github.com/li-xuanhong/source-context-knn-mt

an example in $k$NN-MT is denoted as a key-value pair (Khandelwal et al., 2021), where key is the decoder intermediate representation given the source sentence and the target prefix, and value is the next target token observed in training data. The top-$k$ examples ranked descendingly according to the distance between the keys and the current decoder hidden state are used to estimate the probability distribution to be interpolated. Figure 1 shows the ratio of the top-$k$ examples whose value is equal to the ground-truth target token v.s. the source context distance duing inference, where source context distance is defined as the L2 distance between the representations of the input sentence and the corresponding source sentence of an example (see Sec. 3.1 for more details). It is evident that the ratio roughly inversely correlates to the source context distance, indicating potential usefulness of source context in $k$NN-MT. Moreover, Dai et al. (2023) show that source context is effective in selecting subset of parallel sentence pairs for improving the efficiency of $k$NN-MT. Therefore, source context deserves further attention in $k$NN-MT.

In this work, we revisit the source context in $k$NN-MT and reveal that it is effective in enhancing the three fundamental components of $k$NN-MT, namely, example retrieval, probability distribution estimation and interpolation. By leveraging a dynamic datastore, built with data selected based on source context similarities, we manage to recall additional candidate examples that hold potential utility, yet might be overlooked by the existing $k$NN-MT methods. We then perform a distance calibration of all candidates based on both the source and target contexts, thereby achieving a more precise probability distribution estimation. Additionally, we propose to determine the interpolation coefficient based on the similarity of the source context.

Since an advantage of our proposed method is its seamless integration capability with existing $k$NN-MT methods, we integrate our method with representative $k$NN-MT baselines and conduct experiments on five benchmark datasets from different domains. Experimental results demonstrate that our method consistently outperforms all the baselines across all these datasets, with improvements of up to 1.6 BLEU points. Further ablation studies justify that source context indeed plays a crucial role in all the three fundamental components. Furthermore, our source context-enhanced $k$NN-MT method achieves significant improvements in other sequence-to-sequence tasks that are challenging for existing $k$NN-MT methods, indicating that source context merits further consideration within the $k$NN-MT framework.

## 2 Background

This section briefly introduces $k$NN-MT (Khandelwal et al., 2021), formulated as the following two main steps: datastore construction and prediction with datastore.

**Datastore Construction**  A datastore for $k$NN-MT usually contains decoder intermediate representations and corresponding target tokens as key-value pairs. Specially, given a sentence pair $(\mathbf{x}, \mathbf{y})$ from the training dataset $\mathcal{T} = \{(\mathbf{x}^{(i)}, \mathbf{y}^{(i)})\}$, where $\mathbf{x} = x_1, x_2, \cdots, x_m$ and $\mathbf{y} = y_1, y_2, \cdots, y_n$, it is fed into a pre-trained NMT model $\mathcal{M}$ in a teacher forcing way. The decoder representation at position $t$ is denoted as $f(\mathbf{x}, \mathbf{y}_{<t})$, and the corresponding target token is $y_t$, then the datastore $\mathcal{D}$ is as follows:

$$\mathcal{D} = \{(f(\mathbf{x}, \mathbf{y}_{<t}), y_t) | (\mathbf{x}, \mathbf{y}) \in \mathcal{T}, \forall y_t \in \mathbf{y}\}. \quad (1)$$

**Prediction with Datastore**  Given a source sentence $\mathbf{x}$, the NMT model $\mathcal{M}$ generates the prediction $\hat{\mathbf{y}}$ token by token. Considering the $t$-th time step, the decoder representation is denoted as $\hat{\mathbf{h}}_t = f(\mathbf{x}, \hat{\mathbf{y}}_{<t})$. A $k$NN-MT model uses $\hat{\mathbf{h}}_t$ as a query to retrieve the most similar $k$ entries $\mathcal{N} = \{(\mathbf{h}^{(i)}, v^{(i)})\}_{i=1}^{k}$ in the datastore $\mathcal{D}$. Then the $k$NN distribution $p_{k\text{NN}}(y_t|\mathbf{x}, \hat{\mathbf{y}}_{<t})$ is calculated based on the L2 distances $d(\hat{\mathbf{h}}_t, \mathbf{h}^{(i)})$ between query $\hat{\mathbf{h}}_t$ and the key $\mathbf{h}_i$:

$$p_{k\text{NN}}(y_t|\mathbf{x}, \hat{\mathbf{y}}_{<t}) \propto \quad (2)$$
$$\sum_{(h_i, v_i)} \mathbb{1}_{y_t = v^{(i)}} \exp\left(\frac{-d(\hat{\mathbf{h}}_t, \mathbf{h}^{(i)})}{T}\right),$$

where $T$ is the temperature used to smooth the $k$NN distribution, and the final probability distribution is obtained by interpolating the $k$NN distribution with the NMT distribution:

$$p(y_j|\mathbf{x}, \hat{\mathbf{y}}_{<j}) = \lambda\, p_{k\text{NN}}(y_j|\mathbf{x}, \hat{\mathbf{y}}_{<j})$$
$$+ (1 - \lambda)\, p_{\text{NMT}}(y_j|\mathbf{x}, \hat{\mathbf{y}}_{<j}), \quad (3)$$

where $\lambda$ is the interpolation coefficient, and a larger $\lambda$ means a larger percentage of the $k$NN distribution in the final distribution.

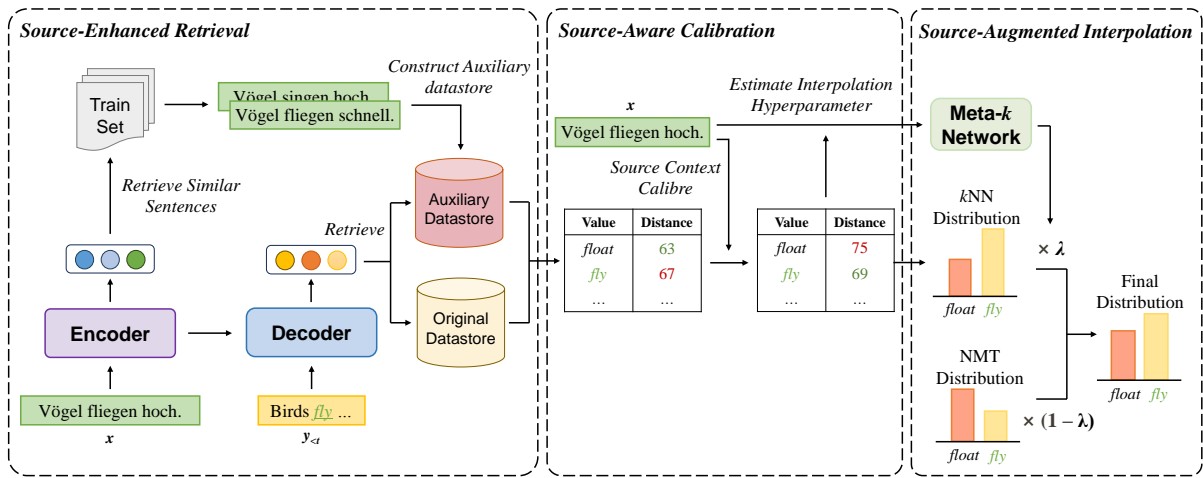

Figure 2: An overview of the proposed method. Source-context enhanced $k$NN-MT from three aspects, namely retrieval, calibration, and interpolation.

## 3 $k$NN-MT via Enhanced Source Context

In this section, we will detail the proposed method, which improves three crucial processes of $k$NN-MT, including retrieval, calibration, and interpolation with source context enhancement. The complete process is illustrated in Figure 2.

### 3.1 Source-Enhanced Retrieval

In the Vanilla $k$NN-MT (Khandelwal et al., 2021), the target-side representation is used as a query to retrieve entries similar to it from the constructed datastore, but the entries retrieved in this way may be affected by noise (Wang et al., 2022a,c) and some desired entries may be missed. Therefore, we adopt an adaptive approach by utilizing text similar to the input text to construct auxiliary datastores. The intuition is that if the source text is similar, there is a higher probability that the target text will also be similar. Consequently, the entries constructed from similar source texts are more likely to be the ground truth than others (Figure 1).

Specifically, we first encode the source texts $\mathbf{x}^{(j)}$ from the training set $\mathcal{T}$ using the encoder of the NMT model $\mathcal{M}$, resulting in the representation $\mathbf{r}^{'(j)} \in \mathbb{R}^{L \times H}$, where $L$ is the sentence length and $H$ is the hidden size, then we take the average over the length dimension to get the final representation $\mathbf{r}^{(j)} \in \mathbb{R}^{H}$. Correspondingly, the input sentence $\mathbf{x}$ to be translated is encoded into $\mathbf{r} \in \mathbb{R}^{H}$.

Then, we calculate the L2 distance $d(\mathbf{r}, \mathbf{r}^{(j)})$ between $\mathbf{x}$ and $\mathbf{x}^{(j)}$ to score the similarity between them. The set of $n$ sentences $\{\mathbf{x}^{(l)}\}_{l=1}^{n}$ with the smallest L2 distance are selected, and their corresponding sentence pairs $\{(\mathbf{x}^{(l)}, \mathbf{y}^{(l)})\}_{l=1}^{n}$ will be

used to construct an auxiliary datastore $\mathcal{D}'$. The construction method is identical to the approach described in Sec. 2. Note that we construct an adaptive auxiliary datastore independently for each input sentence.

Finally, we use $\hat{\mathbf{h}}_t$ as the query to retrieve $k$ nearest entries from each of the datastores $\mathcal{D}$ and $\mathcal{D}'$, and combine them to obtain an entries list whose size is $2k$ for constructing the $k$NN distribution.

### 3.2 Source-Aware Calibration

After source-enhanced $k$NN retrieval, a list of entries $\mathcal{N} = \{(\mathbf{h}^{(i)}, v^{(i)})\}_{i=1}^{2k}$ is retrieved. These entries are now ranked based on their target-side distance $d(\hat{\mathbf{h}}_t, \mathbf{h}^{(i)})$. However, since the NMT model used to compute the keys is not optimized for retrieval, using target-side distance alone may result in suboptimal performance (Wang et al., 2022c). Therefore, we employ source-side distance to calibrate the potentially inaccurate original distance (target-side distance), prioritizing entries that exhibit distance in both the target and source sides. Accordingly, the $k$NN distribution in Eq. 2 is also calibrated.

Specifically, we calculate the sum of the target-side distance $d(\hat{\mathbf{h}}_t, \mathbf{h}^{(i)})$ and the source-side distance $d(\mathbf{r}, \mathbf{r}^{(i)})$ as a new distance metric $d_c$. To better control the weight assigned to the source-side distance in $d_c$, we introduce a hyperparameter $\mu$, and $d_c$ is defined as

$$d_c^{(i)} = d(\hat{\mathbf{h}}_t, \mathbf{h}^{(i)}) + \mu \times d(\mathbf{r}, \mathbf{r}^{(i)}). \quad (4)$$

In order not to affect the speed of inference, we store the target token and its corresponding encoder

output $\mathbf{r}_i$ when constructing the datastore[2]. The entries in the datastore are like $(\mathbf{h}_i, v_i, \mathbf{r}_i)$.

Since the hyperparameter $\mu$ directly affects the performance of the model, setting $\mu$ as a fixed value may hinder the ability of the model to generalize effectively. Therefore, we propose a lightweight module consisting of a two-layer feed-forward network to dynamically adjust the hyperparameter $\mu$ based on the retrieved entries:

$$z = [s^{(1)}, ..., s^{(2k)}; d_c^{(1)}, ..., d_c^{(2k)}; u^{(1)}, ..., u^{(2k)}],$$

$$\mu = \text{sigmoid}(\mathbf{W}_2(\tanh(\mathbf{W}_1 z)) + b_1), \quad (5)$$

where $s^{(i)}$ is source-side distance, $d_c^{(i)}$ is our new distance (Eq. 4), $u^{(i)}$ is the number of unique values in neighbors, and $\mathbf{W}_1$, $\mathbf{W}_2$ are parameter matrices.

The distance metric $d_c$ we propose here considers both the target-side distance and the source-side distance. As the distances of the retrieved entries are calibrated, the $k$NN distriction in Eq. 2 is also calibrated accordingly. Subsequent experimental results further demonstrate that the new distance can effectively mitigate the noise and suboptimality issues caused by the target-only distance.

### 3.3 Source-Augmented Interpolation

Previous work has demonstrated that in the interpolation process of $k$NN-MT, using fixed interpolation coefficients $\lambda$ or retrieval quantities $k$ will decrease the model performance (Zheng et al., 2021a; Jiang et al., 2022) as they tend to treat retrieved information of different quality equally. Inspired by these works, we propose a source-augmented lightweight module to adaptively compute $\lambda$ in the interpolation process:

$$z' = [s^{(1)}, ..., s^{(2k)}; d_c^{(1)}, ..., d_c^{(2k)}; u^{(1)}, ..., u^{(2k)}],$$

$$\lambda = \text{sigmoid}(\mathbf{W}_3(\tanh(\mathbf{W}_4 z')) + b_2), \quad (6)$$

where $\mathbf{W}_3$, $\mathbf{W}_4$ are parameter matrices, and $s^{(i)}$, $d_c^{(i)}$ and $u^{(i)}$ are the same as Eq. 5.

### 3.4 Training Procedure

During training, we freeze the parameters of the NMT model $\mathcal{M}$ and only update the parameters of the lightweight module (like $\mathbf{W}_1$ and $\mathbf{W}_2$), minimizing the cross-entropy loss until convergence.

| Dataset | IT | Medical | Koran | Laws | JRC$_{\text{EnDe}}$ | JRC$_{\text{EnEs}}$ |
|---|---|---|---|---|---|---|
| Train | 222,927 | 248,009 | 17,982 | 467,309 | 699,569 | 679,088 |
| Dev | 2,000 | 2,000 | 2,000 | 2,000 | 2,454 | 2,533 |
| Test | 2,000 | 2,000 | 2,000 | 2,000 | 2,483 | 2,596 |

Table 1: Statistics of datasets.

This ensures that the lightweight modules can dynamically allocate the appropriate $\mu$ and $\lambda$ values based on the quality of the retrieval results.

## 4 Experiments

### 4.1 Experimental Settings

**Datasets and Evaluation** In order to conduct a more comprehensive comparison, we selected datasets separately from the direction of domain adaptation and in-domain translation. For domain adaptation, we follow the same configuration as previous studies (Zheng et al., 2021a; Jiang et al., 2022), conducting experiments on four commonly used benchmark domain datasets (Koehn and Knowles, 2017), including IT, Medical, Koran, and Law. We use the split version from Aharoni and Goldberg (2020) and the translation direction is De→En. For in-domain translation, we perform experiments in four directions on the JRC-Acquis (Steinberger et al., 2006) dataset, including En→De, De→En, En→Es, and Es→En. The statistics of these datasets are shown in Table 1. All data are tokenized by Moses toolkit [3], and we use SacreBLEU [4] to evaluate the performance [5].

**Implementation** We use $k$NN-BOX (Zhu et al., 2023b) to implement multiple $k$NN-MT models. Faiss (Johnson et al., 2021) is employed for entries retrieval. For the four domain datasets, we use WMT 19 winner model (Ng et al., 2019) as the base translation model. For the JRC-Acquis dataset, we train a Transformer (Vaswani et al., 2017) model for each translation direction, which served as the base model. During the training process of the $k$NN-MT model, following the settings of the previous research, we use Adam (Kingma and Ba, 2015) as the optimizer and use the cross-entropy loss for optimization. The learning rate is set to $3 \times 10^{-4}$, and the batch size is 32. The training is stopped when the loss no longer decreases, and hyperparameters are adjusted based on the results from the validation set. The specific model

---

[2]In theory, this approach could require doubling the storage capacity. However, considering that tokens in the same sentence share the same source-side representation, we only utilize an additional 3% of storage space in our implementation.

[3]https://github.com/moses-smt/mosesdecoder
[4]https://github.com/mjpost/sacrebleu
[5]The signature of ScareBLEU is BLEU+case.mixed+numrefs.1+smooth.exp+tok.13a+version.1.5.1

configuration is presented in Appendix A. All experiments are conducted on an Ubuntu server with a RTX3090 GPU.

**Baselines** We use the following models as our baselines:

- Base NMT (Ng et al., 2019): the pre-trained NMT model that is commonly used as the base model of $k$NN-MT.
- Vanilla $k$NN-MT (Khandelwal et al., 2021): the base $k$NN-MT model that introduces the token level nearest neighbor method into NMT for the first time.
- Adaptive $k$NN-MT (Zheng et al., 2021a): the model innovatively introduces a lightweight trainable module to adaptively determine hyperparameters such as $k$ and $\mu$ during distribution interpolation.
- KSTAR (Jiang et al., 2021): the model that adds a learnable kernel to dynamically calculate the similarity of retrieved instances and smooth the obtained $k$NN distribution.
- CLKNN (Wang et al., 2022c): the model that enhances the distinctiveness of the representation of keys in the datastore by employing supervised contrastive learning.
- Robust $k$NN-MT (Jiang et al., 2022): the model that calibrates the $k$NN distribution and adds noise training to enhance the robustness.

## 4.2 Main Results

**Domain Adaptation** The main results of the four domain datasets are shown in Table 2, where the performance of the strong baseline before and after the enhancement of our method is listed. Note that due to Adaptive $k$NN-MT simultaneous adaptively determine both $\lambda$ and $k$, when enhancing Adaptive $k$NN-MT using our method, we additionally incorporate the source-side distance as the input to dynamically determine $k$. This differs from the approach described in Sec. 3.3, which focuses on adaptively calculating $\lambda$. The same rationale applies to Robust $k$NN-MT enhancement as well.

It can be found that after being enhanced by our method, two strong $k$NN-MT baselines (Zheng et al., 2021a; Jiang et al., 2022) have gained consistent and significant performance improvement on all datasets. For example, the enhanced Adaptive $k$NN-MT achieves an average increase of 1.57 BLEU points (47.74 to 49.31) on the IT dataset,

surpassing the strong baseline Robust $k$NN-MT (48.78). Furthermore, after enhancement, Robust $k$NN-MT can achieve a higher BLEU score (49.66) on this dataset. Through an overall analysis of the four datasets, our method demonstrates a significant improvement, with an average increase of 1.1 and 0.7 BLEU scores for Adaptive $k$NN-MT and Robust $k$NN-MT, respectively. These results demonstrate the effectiveness and the generality of our method, as it can be applied to enhance many $k$NN-MT models and bring consistent performance improvements.

A question worth contemplating is why the enhancement effect on the Koran dataset is not as significant as on other datasets, and the notable performance of the CLKNN (Wang et al., 2022c) on this dataset provides some insights. By employing supervised contrastive learning, CLKNN enhances the distinctiveness of the representations of keys in the datastore. However, we still utilize the outputs of the original NMT model as keys. Considering the sparsity of correct entries in such small datasets, retrieving the correct entries becomes even more challenging when using these indistinct keys for retrieval. Conversely, for larger-scale datasets, the abundance of correct entries may make it easier to retrieve the ground truth, and it can be observed that our enhancement method performs well on large-scale datasets.

**In-Domain Translation** The results on four domain datasets demonstrate the ability of our method to enhance the domain knowledge transfer of $k$NN-MT. Furthermore, we aim to demonstrate that the $k$NN-MT model can enhance the utilization of in-domain knowledge through token-level information retrieval, thereby mitigating issues such as knowledge forgetting, and our proposed method further strengthens this ability. To achieve this, in contrast to using pre-trained NMT models on domain datasets, we utilized NMT models trained on the source domain specifically for the JRC-Acquis dataset.

The results in Table 3 indicate that the $k$NN-MT model can effectively enhance the original NMT model by retrieving and utilizing in-domain knowledge but still lags behind several Translation Memory (TM) base strong baselines. After being enhanced by our method, the performance of $k$NN-MT has been significantly improved. For example, the Adaptive $k$NN-MT and Robust $k$NN-MT achieve 1.5 and 0.8 average BLEU improvement

| Model | IT | Medical | Koran | Laws | Avg |
|---|---|---|---|---|---|
| Base-NMT | 38.35 | 40.06 | 16.26 | 45.48 | 35.03 |
| Vanilla $k$NN-MT | 45.72 | 54.26 | 20.29 | 61.27 | 45.38 |
| KSTAR | 47.72 | 56.81 | 19.97 | 63.34 | 46.96 |
| CLKNN[*] | 47.84 | 55.87 | **21.81** | 62.01 | 46.88 |
| Adaptive $k$NN-MT | 47.74 | 56.12 | 20.31 | 62.87 | 46.76 |
| + Ours | 49.31[†] (+1.57) | 57.72[†] (+1.60) | 20.38 (+0.07) | 64.14[†] (+1.27) | 47.89 (+1.13) |
| Robust $k$NN-MT | 48.78 | 57.11 | 20.50 | 63.61 | 47.50 |
| + Ours | **49.66**[†] (+0.88) | **57.78**[†] (+0.67) | 20.72 (+0.22) | **64.52**[†] (+0.91) | **48.17** (+0.67) |

Table 2: The BLEU scores on test sets of four domain benchmarks. "*" indicates the results from original papers, and "†" denotes a statistically significant improvement ($p < 0.05$).

| Model | Es→En | En→Es | De→En | En→De | Avg |
|---|---|---|---|---|---|
| (Zhang et al., 2018)[★] | 64.30 | 61.56 | 60.26 | 55.14 | 60.31 |
| (Xia et al., 2019)[★] | 66.21 | 62.76 | 61.72 | 56.88 | 61.89 |
| (Cai et al., 2021)[★] | 66.48 | 62.76 | 63.85 | 57.53 | 62.65 |
| (Cheng et al., 2022a)[★] | 67.76 | 64.04 | 64.33 | 58.69 | 63.70 |
| Transformer | 62.71 | 60.54 | 58.93 | 53.32 | 58.87 |
| Vanilla $k$NN-MT | 65.13 | 62.78 | 63.03 | 54.08 | 61.25 |
| Adaptive $k$NN-MT | 66.01 | 64.09 | 64.59 | 58.54 | 63.30 |
| + Ours | 67.87[†] (+1.86) | 65.53[†] (+1.44) | 65.67[†] (+1.08) | 60.31[†] (+1.47) | 64.84 (+1.54) |
| Robust $k$NN-MT | 67.08 | 64.97 | 64.93 | 59.52 | 64.13 |
| + Ours | **67.95**[†] (+0.87) | **65.60**[†] (+0.63) | **65.73**[†] (+0.80) | **60.48**[†] (+0.96) | **64.94** (+0.81) |

Table 3: The BLEU scores on test sets of JRC-Acquis. "★" highlights results provided by Cheng et al. (2022b), and "†" marks an improvement that is statistically significant ($p < 0.05$).

| Adaptive $k$NN-MT | 47.74 |
|---|---|
| *w/* Retrieval | 48.08 (+0.34) |
| *w/* Calibration | 48.63 (+0.89) |
| *w/* Interpolation | 48.25 (+0.51) |
| *w/* All | **48.89** (+1.15) |

Table 4: The impact of each enhanced component on the overall performance.

in four directions of JRC-Acquis and successfully surpassing these TM-base strong baselines, thus demonstrating the effectiveness of our source-side enhancement.

## 4.3 Ablation Study

Our proposed method enhances three $k$NN-MT processes (retrieval, calibration, and interpolation). To investigate the utility of each enhanced module, we conducted ablation experiments on the IT dataset. Specially, we individually add each enhanced component and observed the performance changes. The results are shown in Table 4.

It can be observed that the performance of the model has improved after incorporating each enhancement module, particularly with the most significant improvement seen after the inclusion of

the calibration module. This is because the model can only rely on target-side distances to determine the weights of retrieved entries without the source-aware calibration module, which makes the model susceptible to noise and leads to suboptimal performance. This finding further highlights the importance of a comprehensive consideration of the source-side context in $k$NN-MT.

It should be noted that, for a fair comparison, we set the value of $k$ to that which performs optimally in Adaptive $k$NN-MT ($k = 8$), rather than the value most suited for our approach. As a result, the value of the "w/ All" term in Table 4 (48.89) does not attain our highest reported performance (49.31). Consequently, owing to the constraints imposed by a smaller $k$ value, the retrieval and interpolation modules, despite exhibiting performance improvements, have not been able to fully manifest their capabilities.

## 5 Analysis

In this section, we attempt to explore several important and meaningful questions through experimental analysis.

- **Q1**: Does the model enhanced with our

method demonstrate robustness when setting different values of $k$? (Sec. 5.1)

- **Q2**: Why does our method deliver performance gains? (Sec. 5.2)

- **Q3**: Can our proposed enhanced method be applied to other text generation tasks besides translation? (Sec. 5.3)

## 5.1 Robustness (Q1)

Previous studies have found that the Vanilla $k$NN-MT model is highly sensitive to the setting of the retrieval entries number $k$, often achieving optimal performance only when $k$ is set to a moderate value. This may limit the capacity of retrieval information that $k$NN-MT can utilize, resulting in suboptimal performance. Some research enhanced the robustness of the $k$NN-MT model through supervised training to adaptively determine hyperparameters (Zheng et al., 2021a) or calibrate the $k$NN distribution (Jiang et al., 2022).

To demonstrate that the model enhanced by our method can achieve better robustness while improving performance, following previous studies, we conducted experiments on the IT dataset, considering different settings of $k$. Note that our enhanced model retrieves the same number of entries from both the original datastore and the auxiliary datastore. To ensure a fair comparison, in this section, when we mention the enhanced model retrieves $k$ entries, it refers to the following scenario: the model retrieves $k/2$ entries from both the original and the auxiliary datastore and concatenates them to form a list of size $k$. The experimental results are shown in Figure 3.

We can observe that across all value settings of $k$, the model enhanced by the proposed enhanced method consistently surpasses the original model, demonstrating the effectiveness of our method. Moreover, the enhanced model can achieve robust performance as $k$ increases. For example, the enhanced Adaptive $k$NN-MT model maintains a high-performance improvement as $k$ increases, while the original model slowly declines after $k = 8$. This suggests that our method can effectively filter out noise in retrieval entries and appropriately utilize the relevant knowledge to enhance the translation model.

Another key issue is the inference speed. Theoretically, given the same value of $k$, calibration and interpolation introduce only a marginal increase in computation, while the retrieval computational

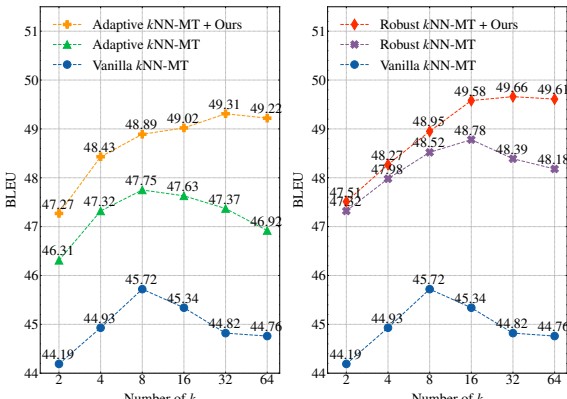

Figure 3: BLEU scores v.s. $k$ on the IT dataset.

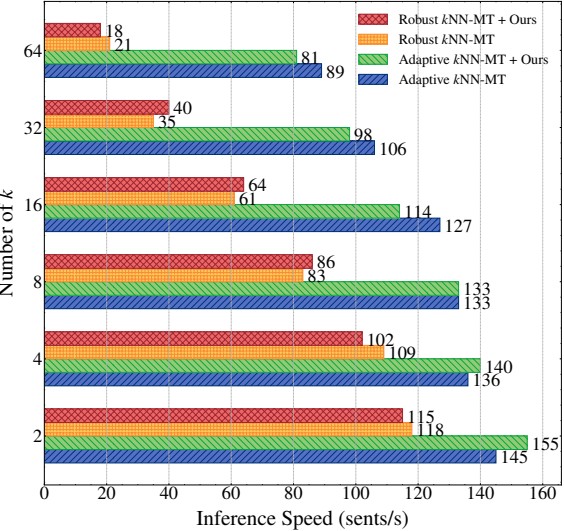

Figure 4: Inference speed v.s. $k$ on the IT dataset.

cost remains roughly unchanged. As a result, the overall additional computational overhead is minimal. In Figure 4, we present the inference speed of various $k$NN-MT models and their corresponding enhanced models. We report the average speed of five trials for each setting. It can be observed that the enhanced model and the original model have comparable inference speeds, which justifies the theoretical analysis. Therefore, our proposed method achieves sustained and stable performance growth at a remarkably low cost.

## 5.2 Accuracy (Q2)

To investigate the source of performance gains due to the source-side enhancement method, we analyze it from the perspective of retrieval accuracy.

A retrieved entry is considered correctly retrieved if its value matched the ground-truth token. The experiments are performed on the validation set of the IT dataset. Several metrics like P@n

| Model | P@1 | P@5 | P@20 | MAP@5 | MAP@20 | BLEU |
|---|---|---|---|---|---|---|
| Vanilla $k$NN-MT | 0.3382 | 0.3176 | 0.2968 | 0.3262 | 0.3097 | 42.17 |
| Adaptive $k$NN-MT | 0.3408 | 0.3188 | 0.2971 | 0.3279 | 0.3106 | 43.98 |
| + Ours | 0.3504 | 0.3405 | 0.3132 | 0.3492 | 0.3296 | 46.43 |
| Robust $k$NN-MT | 0.3496 | 0.3267 | 0.3038 | 0.3365 | 0.3181 | 45.15 |
| + Ours | **0.3621** | **0.3433** | **0.3174** | **0.3523** | **0.3334** | **46.72** |

Table 5: The accuracy of retrieved entries of different $k$NN-MT models on IT validation set.

and MAP@n (the average of AP@n scores across all queries) are employed to evaluate retrieval accuracy. The results are shown in Table 5. It can be observed that the enhanced model surpasses the original model in all retrieval metrics, demonstrating that our method enables the $k$NN-MT to retrieve more correct entries, thereby leading to better translation results.

An interesting observation is that Adaptive $k$NN-MT, compared to Vanilla $k$NN-MT, shows minimal improvement in retrieval metrics but a significant increase in the BLEU score. This might be attributed to the content of the Adaptive $k$NN-MT, which enhances performance by adaptively adopting retrieval entries, and this improvement is not related to retrieval accuracy. However, after our method enhancement, Adaptive $k$NN-MT's retrieval accuracy is significantly improved and the performance is further enhanced. Combined with our proposed method like improved calibration distance, we suspect that the performance improvement is likely due to enhanced retrieval accuracy.

### 5.3 Generality (Q3)

To validate the generality of our proposed method, we conducted experiments on several text generation tasks other than translation. We collect task-oriented dialogue generation and question generation datasets from (Tang et al., 2022). Since these datasets did not provide testing set, we split the training set into a new training set and a validation set in a $9 : 1$ ratio and report the experimental results on the original validation set. We trained a Transformer model as the base model on each dataset. The BLEU metric is used to evaluate the performance of the models, and the results are reported in Table 6.

Based on the results presented in Table 6, it can be observed that the $k$NN-MT method effectively enhances the performance of the model on these text generation datasets, indicating the potential of the $k$NN-MT method in non-translation text gen-

| Model | DG | QG |
|---|---|---|
| Transformer | 39.86 | 29.56 |
| Vanilla $k$NN-MT | 41.09 | 30.85 |
| Adaptive $k$NN-MT | 41.67 | 31.16 |
| +Ours | 43.18 (+1.51) | 32.09 (+0.93) |
| Robust $k$NN-MT | 42.64 | 31.48 |
| +Ours | **43.51** (+0.87) | **32.26** (+0.78) |

Table 6: The experimental results on dialogue generation (DG) and question generation (QG) tasks.

eration tasks. Additionally, by enhancing with our method, the performance of these $k$NN-MT models can be further improved, thus demonstrating the effectiveness and generality of our model.

## 6 Related Work

Existing approaches commonly improve $k$NN-MT (Khandelwal et al., 2021) from effectiveness and efficiency perspectives.

**Effectiveness** To enhance the effectiveness of $k$NN-MT, Zheng et al. (2021a) adaptively determine the retrieved entries to be utilized. Jiang et al. (2021) incorporates kernel smoothing methods to $k$NN-MT. Jiang et al. (2022) calibrate $k$NN distribution and make it more robust by noise-resistant training. Wang et al. (2022c) employs contrastive learning to make the keys of entries more discriminative to improve retrieval accuracy, while Li et al. (2022) utilizes representations from pre-trained models to construct a higher-quality datastore. It can be observed that these works lack explicit consideration of the source context. In contrast, we incorporate source context in all three processes of $k$NN-MT: retrieval, calibration, and interpolation, resulting in consistent performance improvements.

**Efficiency** Constrained by the performance requirements imposed by large datastore for inference, researchers explore approaches to enhance the efficiency of $k$NN-MT. Wang et al. (2022a) reduces the dimensionality of entry features and prunes redundant entries by clustering. Zhu et al.

(2023a) explores the knowledge required by $k$NN-MT and filter out redundant entries according local correctness. Deguchi et al. (2023) retrieves tokens from neighbor sentences instead of all sentences and using a lookup table for distance calculation. Dai et al. (2023) constructs a mini-datastore using sentences similar to the input during each inference instead of the original datastore and inspired us to introduce the auxiliary datastore in our work. As shown in Figure 4, our method introduces limited extra computational cost, achieving a good balance between effectiveness and efficiency.

## 7 Conclusion

In this paper, we revisit the role of source context and propose a novel method for improving neural machine translation via source context enhancement, which augments the key processes of $k$NN-MT (i.e., retrieval, calibration, and interpolation). Experimental results demonstrate that it can simply and effectively enhance most mainstream $k$NN-MT models, achieving consistent improvements. Moreover, our method can significantly improve the robustness of the $k$NN-MT model and help improve its effectiveness in dialogue generation and question generation. We look forward to further exploring the practical utility and efficiency of this method in more natural language generation tasks in the future.

## Limitation

While our method has exhibited effectiveness, its exploration beyond machine translation in terms of generality has been confined to just two additional text generation datasets, namely dialogue generation and question generation. Consequently, it is imperative to investigate our method across a more diverse set of tasks. Existing research (Dai et al., 2023) indicates that the improvement brought by $k$NN-MT on certain large-scale translation datasets are not sufficiently significant. In future work, we aim to rigorously assess the effectiveness of our method on these large-scale translation datasets. Although our work achieves a good balance between effectiveness and efficiency, we believe there still room for improving efficiency without compromising effectiveness by leveraging source context, which will be explored in the future.

## Acknowledgement

This work is supported by the National Social Science Fund of China under Grant No. 20BTQ068. We thank all anonymous reviewers for their insightful suggestions on this work.

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

## A    Model Configuration

The specific hyperparameter settings of the Transformer model trained on the JRC-Acquis dataset are presented in Table 7.

| Hyperparameter | Value |
| --- | --- |
| Embedding Dim | 512 |
| Attention Head | 8 |
| Encoder Layer | 6 |
| Decoder Layer | 6 |
| Dropout | 0.1 |
| Hidden Dim | 512 |

Table 7: The configuration of our Transformer model.

## B    Datastore

Following previous work (Zheng et al., 2021a), we employ numpy.array [6] to store key-value pairs and utilized faiss [7] for index construction. The cluster centroids are set to $4,096$ and $32$ during training and retrieval, respectively, while the vector code size is set to $64$. The specific disk space occupied is presented in Table 8. It can be observed that the storage occupied by the source context is about 3% of the total original storage, demonstrating the low space requirement of our work.

| Datastore | IT | Medical | Koran | Law |
| --- | --- | --- | --- | --- |
| Key-Value | 6.93 | 13.22 | 1.00 | 36.51 |
| Source Context | 0.42 | 0.47 | 0.05 | 0.89 |

Table 8: Disk space (GB) occupied by different datastores.

## C    Experiments on Larger Datasets

Following the settings in recent strong baselines (Zheng et al., 2021a; Jiang et al., 2022), we conduct experiments on the four datasets, i.e., IT, Medical, Koran, and Law, to make our work directly comparable with these recent strong baselines. We also conduct experiments on the Subtitles dataset (Koehn and Knowles, 2017), whose training set consists of about 14 million parallel sentence pairs, two orders of magnitude larger than the previous four datasets. The translation direction is De→En. For domain adaptation setting, we use the WMT19 De→En direction winner model (Ng et al., 2019) as the base model. For in domain setting, we trained a Transformer (Vaswani et al.,

---

2017) model from scratch using the Subtitle training set as the base model. The experimental results are reported in Table 9. We can see that the two strong $k$NN-MT baselines remain effective on this larger dataset. Moreover, our method introduces significant improvements under both the Domain Adaptation and In Domain settings.

| Model | DA | ID |
| --- | --- | --- |
| Base Model | 29.58 | 29.96 |
| Adaptive $k$NN-MT | 31.23 | 31.38 |
| + Ours | 32.08$^\dagger$ (+0.85) | 32.47$^\dagger$ (+1.09) |
| Robust $k$NN-MT | 31.68 | 31.91 |
| + Ours | **32.39**$^\dagger$ (+0.71) | **32.66**$^\dagger$ (+0.75) |

Table 9: The BLEU scores for the Subtitle dataset test set in the Domain Adaptation (DA) and In Domain (ID) settings. "$\dagger$" denotes a statistically significant improvement ($p < 0.05$).

## D    Auxiliary Datastore

Note that the auxiliary datastore is constructed from the top $n$ similar sentence pairs ranked by the source-side similarity between the input sentence and the source side of the sentence pairs. Therefore, we will recall different entries from the original and auxiliary datastores even if the same query is used. Experiments on the development set of the IT dataset reveal that only 7.38% of tokens retrieved from the auxiliary datastore overlap with those from the original datastore. Here, "overlap" means the two tokens are identical and from the same sentence pairs. This result indicates that the auxiliary datastore has the potential to (1) introduce more unique tokens and (2) recall complementary references for the same tokens. In conjunction with our ablation study results on the retrieval module, this underscores the importance of considering source context when constructing datastore.

Another noteworthy issue is that the size of the auxiliary datastore is about 4-5 orders of magnitude smaller than the size of the original datastore, for instance, 285 v.s. 3,602,862 on the IT dataset. This means that the construction and retrieval of the auxiliary datastore necessitate minimal computational expenditure.

---

[6] https://numpy.org
[7] https://github.com/facebookresearch/faiss