# OpenReview forum: "Revisiting Source Context in Nearest Neighbor Machine Translation"
_EMNLP/2023/Conference — EMNLP 2023 Main_

### Official Review · Reviewer_4RbX · 2023-07-28

**Soundness:** 3

**Excitement:**

3: Ambivalent: It has merits (e.g., it reports state-of-the-art results, the idea is nice), but there are key weaknesses (e.g., it describes incremental work), and it can significantly benefit from another round of revision. However, I won't object to accepting it if my co-reviewers champion it.

**Missing References:**

NA

**Paper Topic And Main Contributions:**

This paper investigates the role of source context in kNN-MT. The authors argue that source context warrants further attention, since it plays an important role in kNN-MT but is mostly overlooked by earlier work.

The authors propose methods to incorporate source context in multiple kNN-MT components:
* Retrieval. An auxiliary datastore based on similarity with current input sentence is created, and used in conjunction with the vanilla datastore.
* Calibration. Instead of relying solely on target side distance, the method also takes into account source side distance with the goal of removing noisy matches.
* Interpolation. \lambda is computed adaptively based on source similarity (amongst other factors).

Results seem to improve over strong baselines, while inference speed and memory requirements only marginally increase.

**Questions For The Authors:**

A. Can you clarify details regarding evaluation?

B. Can you clarify details regarding method?

**Reasons To Accept:**

* Using source context for kNN-MT makes a lot of sense, but is an under-explored direction. The authors provide good motivation for this direction, and results seem promising.
* The method is quite comprehensive. Source context is used in multiple kNN-MT components, and an ablation indicates that it brings benefits to every component, and more so when combined.
* Inference speed and memory requirements are only marginally increased compared to earlier work.
* The paper is well-written and easy to follow, thanks to clear structuring and a clear overview figure.

**Reasons To Reject:**

* Evaluation misses important details

The SacreBLEU signature is missing, so it is not clear whether the comparison with the baselines is fair. From the text I understand that for all models reported in Tables 2 and 3, except Transformer* and your own models, you report scores from the original papers? But these scores do not match the numbers reported in the papers. For instance, the numbers reported for adaptive kNN-MT do not match the numbers in the original adaptive kNN-MT paper.

Without additional information and context, it is hard to draw any conclusions from these numbers.

* Some important method details are missing.

It is unclear how distance d(r,r^j) (line 173) between x and x^j is calculated, given that x and x^j in general do not have the same length. Similar for d(h_t, h^i) (line 202).

It is unclear how the lightweight module that adjusts hyperparameter \mu is optimized/learned. In particular, how are the weights of W_1, W_2, and b_1 learned? Similar for equation 6.

\lambda is computed adaptively for your method. what about other hyperparameters such as number of neighbors k and temperature T, are they found using a hyperparameter search? If so, which values do you search over? What about for the other methods you compare to, which sets do you use there? Without this information, it is not possible to determine the amount of hyperparameter tuning for your method vs the baselines.

**Reproducibility:**

3: Could reproduce the results with some difficulty. The settings of parameters are underspecified or subjectively determined; the training/evaluation data are not widely available.

**Reviewer Confidence:**

5: Positive that my evaluation is correct. I read the paper very carefully and I am very familiar with related work.

**Typos Grammar Style And Presentation Improvements:**

* Line 123: "it is feed into" -> "it is fed into"
* Line 135: "is calculates" -> "is calculated"
* Line 168-171: it's confusing to use x^{(j)} and x to both denote a source sentence.
* Line 240: "these work" -> "these works"
* Line 248: "Experiment" -> Experiments
* Line 388: "To deeply investigate" -> "to investigate"
* Line 420-427: it would be clearer to choose better names for the subsections instead of this bulleted list. E.g., "5.1 How robust is the model for different values of k?". Similarly for the other subsections. Then the bulleted list is not necessary anymore and can be removed.
* Line 462-463: "This proves" -> "This suggests" (an empirical observation is not a proof.)
* Line 751: "employe" -> "employ"

---

> ### Author Rebuttal · Authors · 2023-08-29
>
> **Question A: Fairness of the comparisons**
>
> **Response A:**
>
> - We find the reviewer has big concern on the fairness of the comparisons in our paper. **We believe that they are indeed fair.** We will delve into the specifics in the subsequent paragraphs.
>
> - **Evaluation Metric is Aligned**
>
>   - The signature of ScareBLEU is `BLEU+case.mixed+numrefs.1+smooth.exp+tok.13a+version.1.5.1`, which is the same as the baselines.
>
> - **Baseline Results can be Compared Directly:**
>
>   - **Domain Adaptation (Refer to Table 2 in the original submission):** Apart from CLKNN, the results for all other models come from our implementation. Notably, the original KSTAR paper trained the KSTAR model using Transformer, we implement it with the WMT19 winner model[1] to ensure a fair comparison, which might explain the improved performance over the original paper. The subsequent table show the differences between our implement results and the original papers.
>
>     | Model                                                   | IT    | Medical | Koran | Law   | Avg   |
>     | ------------------------------------------------------- | ----- | ------- | ----- | ----- | ----- |
>     | Base NMT                                                | 37.98 | 39.91   | 16.30 | 45.71 | 34.97 |
>     | Base NMT (Our Implementation)                           | 38.35 | 40.06   | 16.26 | 45.48 | 35.03 |
>     |                                                         |       |         |       |       |       |
>     | Vanilla kNN-MT                                          | 45.82 | 54.35   | 19.45 | 61.78 | 45.35 |
>     | Vanilla kNN-MT (Our Implementation)                     | 45.72 | 54.26   | 20.29 | 61.27 | 45.38 |
>     |                                                         |       |         |       |       |       |
>     | KSTAR                                                   | 35.74 | 53.40   | 16.97 | 59.41 | 41.38 |
>     | KSTAR (Our Implementation)                              | 47.72 | 56.81   | 19.97 | 63.34 | 46.96 |
>     |                                                         |       |         |       |       |       |
>     | Adaptive kNN-MT[2]                                      | 48.04 | 56.41   | 21.09 | 63.21 | 47.18 |
>     | Adaptive kNN-MT (Robust kNN-MT Author's Implementation) | 47.88 | 56.10   | 20.43 | 63.20 | 46.90 |
>     | Adaptive kNN-MT (INK Author's Implementation)           | 47.37 | 56.21   | 20.44 | 63.13 | 46.79 |
>     | Adaptive kNN-MT (Our Implementation)                    | 47.74 | 56.12   | 20.31 | 62.87 | 46.76 |
>     |                                                         |       |         |       |       |       |
>     | Robust kNN-MT[3]                                        | 48.90 | 57.28   | 20.71 | 64.07 | 47.74 |
>     | Robust kNN-MT (INK Author's Implementation)             | 48.50 | 57.12   | 20.81 | 63.74 | 47.54 |
>     | Robust kNN-MT (Our Implementation)                      | 48.78 | 57.11   | 20.50 | 63.61 | 47.50 |
>
>   - It can be seen the average BLEU score difference between our results and the original paper across the four datasets is within 0.2 BLEU score. Except for Adaptive kNN-MT, for which we additionally cited Robust kNN-MT[3] and INK[4] for their implementation results. Their results are similar to ours. For example, INK's implementation average BLEU is 46.79, and ours is 46.76.
>
>   - **In Domain (Refer to Table 3 in the original submission):** The first four baselines reused the results from Cheng et al.[5]. The other methods, including Transformer, Vanilla kNN-MT, Adaptive kNN-MT, Adaptive kNN-MT + Ours, Robust kNN-MT, and Robust kNN-MT + Ours, were implemented by us. We intend to showcase the performance improvement our method can bring by comparing model with or without our method. It can be seen that under the same implementation conditions, Adaptive kNN-MT + Ours has an average increase of 1.54 BLEU in four directions, and Robust kNN-MT + Ours has an average increase of 0.81 BLEU, showing the effectiveness of our proposed method.
>
> - **Hyperparameters for the Baselines are also Optimized:**
>
>   - For Adaptive kNN-MT, the best values for $k$ and $T$ were determined through grid search on the development set. For Adaptive kNN-MT + Ours, the best values for $k$ and $T$ were also determined through grid search on the development set.
>   - For Robust kNN-MT, the best value for $k$ was determined through search on the development set ($T$  value is dynamically calculated). For Robust kNN-MT + Ours, the best value for $k$ was determined through search on the development set ($T$ value is dynamically calculated).
>
> - Therefore, we are confident that our comparisons are fair, the results are trustworthy, and the observed improvements are substantial.
>
> **Question B: Can you clarify details regarding method (how to calculate the distance between representations r in the auxiliary datastore, how the lightweight module is trained)?**
>
> **Response B:**
>
> - **Distance Computation between Representations in Auxiliary Datastore:** We encode the source sentence $x$ using the NMT model's Encoder module to obtain a tensor of size ($L$, $H$), where $L$ is the sentence length, and H is the hidden size. We average over the length dimension to get a representation $r$ of size ($1$, $H$), which serves as the context representation of the source sentence. Similarly, the retrieved sentence can also be transformed into a representation $r^j$ of size ($1$, $H$) in a similar manner. We then calculate the L2 distance between them to obtain similarity.
> - **Training of the Lightweight Module:** During training, we freeze the parameters of the NMT model and only update the parameters of the lightweight module ($W_1$, $W_2$, and $b_1$), minimizing the cross-entropy loss until convergence. This ensures that the lightweight module can dynamically allocate the appropriate lambda value based on the quality of the retrieval results.
> - We will add these details to the revised version. Thank you for the suggestion.
>
> **Question C: Grammatical errors and presentation improvements.**
>
> **Response C:**
>
> - Thank you for your detailed suggestions. We will fix them in the revised version.
>
> &nbsp;
>
> [1] Ng N, Yee K, Baevski A, et al. Facebook FAIR’s WMT19 News Translation Task Submission[C]//Proceedings of the Fourth Conference on Machine Translation (Volume 2: Shared Task Papers, Day 1). 2019: 314-319.
>
> [2] Zheng X, Zhang Z, Guo J, et al. Adaptive Nearest Neighbor Machine Translation[C]//Proceedings of the 59th Annual Meeting of the Association for Computational Linguistics and the 11th International Joint Conference on Natural Language Processing (Volume 2: Short Papers). 2021: 368-374.
>
> [3] Jiang H, Lu Z, Meng F, et al. Towards Robust k-Nearest-Neighbor Machine Translation[C]//Proceedings of the 2022 Conference on Empirical Methods in Natural Language Processing. 2022: 5468-5477.
>
> [4] Zhu W, Xu J, Huang S, et al. INK: Injecting kNN Knowledge in Nearest Neighbor Machine Translation[J]//Proceedings of the 2023 Annual Meeting of the Association for Computational Linguistics. 2023: 15948–15959.
>
> [5] Cheng X, Gao S, Liu L, et al. Neural Machine Translation with Contrastive Translation Memories[C]//Proceedings of the 2022 Conference on Empirical Methods in Natural Language Processing. 2022: 3591-3601.

---

### Official Review · Reviewer_qmbQ · 2023-08-04

**Soundness:** 4

**Excitement:**

4: Strong: This paper deepens the understanding of some phenomenon or lowers the barriers to an existing research direction.

**Missing References:**

- It might be nice to cite this concurrent work, which uses source-side information to filter the datastore. https://aclanthology.org/2023.acl-long.10/

**Paper Topic And Main Contributions:**

The conventional kNN-MT only uses decoder-side information to retrieve the kNN tokens from the datastore. However, the source-side context should be considered as well.
Therefore, this paper proposes a new method to incorporate the source-side contexts into kNN-MT.

The proposed method comprises source-enhanced retrieval, source-aware calibration, and source-augmented interpolation.
The source-enhanced retrieval adds a new auxiliary datastore, constructed and searched by encoder-side information.
During the inference, the model uses the conventional decoder-based and the new encoder-based auxiliary datastore.
The source-aware calibration modifies the distance function, which is used to score the retrieved kNN tokens, to consider source-side distance.
The source-augmented interpolation adaptively computes $\lambda$, which weighs the kNN tokens and original NMT scores.

They conducted experiments mainly on machine translation tasks. They confirmed that the new source-context aware module can combine with several kNN-MT models and consistently improves both the domain adaptation and in-domain translation tasks.
Ablation study shows the calibration module most contributes to the accuracy.

Analysis reveals how the new method is robust to different hyper-parameter settings, comparison of inference speed, retrieved tokens accuracy, and the results on the other two tasks.


**Questions For The Authors:**

- A.  In line 170, which encoder state is used to create the datastore? Is it a sentence or token-level representation? If it is a sentence-level representation, how can you generate it from encoder states? If it is token-level, which encoder state do you use as a query during inference?
- B. In line 223, what does "the number of unique values in neighbors" mean? Does it mean unique tokens?
- C. In line 489, please clarify or cite how P@n and MAP@n are calculated.
- D. Have you conducted experiments with distant language pair such as English-Chinese, or English-Japanese? It would be nice to see whether the proposed method works well even if the source and target representation can be largely different.
- E. How do the tokens retrieved from the auxiliary datastore and original datastore overlap? I'm curious how unique tokens can be retrieved by using the auxiliary datastore.
- F. I understand that your method constructs the auxiliary datastore for each sentence. How heavy is it to construct?

**Reasons To Accept:**

- The proposed source-context aware kNN-MT method is exciting and consistently improves performance on machine translation, dialogue generation, and question generation tasks.
- The proposed method also improves the robustness of the number of retrieved tokens (k).
- The comparison of inference speed shows the proposed method does not drastically slow the inference, which is essential for practical use.

**Reasons To Reject:**

- Some details of the proposed method are unclear to me, and it might make it difficult to reproduce. See Questions for more details.


**Reproducibility:**

3: Could reproduce the results with some difficulty. The settings of parameters are underspecified or subjectively determined; the training/evaluation data are not widely available.

**Reviewer Confidence:**

4: Quite sure. I tried to check the important points carefully. It's unlikely, though conceivable, that I missed something that should affect my ratings.

---

> ### Author Rebuttal · Authors · 2023-08-29
>
> **Question A: In line 170, which encoder state is used to create the datastore? Is it a sentence or token-level representation? If it is a sentence-level representation, how can you generate it from encoder states? If it is token-level, which encoder state do you use as a query during inference?**
>
> **Response A:**
>
> - In our implementation, we use the encoder of the NMT model to encode the source sentence, resulting in a tensor of size ($L$, $H$), where L is the sentence length and $H$ is the hidden size. We take the average over the length dimension to get a representation of size ($1$, $H$) denoted as $r$. This serves as the context representation of the source sentence and is used as a query. We will provide more details in the revised version.
>
> **Question B: In line 223, what does "the number of unique values in neighbors" mean? Does it mean unique tokens?**
>
> **Response B:**
>
> - This concept comes from a previous work Adaptive KNN-MT[1]. It refers to the number of unique tokens. For example, if $k$=4, and the retrieved set of entries is (203, 203, 203, 837), the unique tokens are 203 and 837. Another set of entries, (203, 203, 508, 837), would have unique tokens 203, 508, and 837. Previous research[1] found that unique tokens affect confidence calculations. We will elaborate on the revised version.
>
> **Question C: In line 489, please clarify or cite how P@n and MAP@n are calculated.**
>
> **Response C:**
>
> - P@n refers to precision at $n$, which is widely used in information retrieval[2]. Suppose there are $c$ ground truth tokens in the top $n$ retrieved entries, then we have $P@n=\frac{c}{n}$. And the average value of P@n over the entire evaluation dataset is reported in our paper.
> - AP@n is defined as $AP@n = \frac{1}{NF} \sum_{i=1}^{n} \operatorname{rel}(i) \times P@i$, where $rel(i)$ is an indicator function that equals 1 when the $i$-th retrieved entry is a ground truth token, and 0 otherwise, and $NF$ is the total number of ground truth tokens in the top $n$ retrieved entries[3]. MAP@n is defined as the mean of AP@n values across $Q$ retrievals[4]: $MAP@n = \frac{1}{Q} \sum_{q=1}^{Q} (AP@n)_q$
>
> - Thanks for your suggestion. We will provide more details in the revised version.
>
> **Question D: Have you conducted experiments with distant language pairs such as English-Chinese, or English-Japanese? It would be nice to see whether the proposed method works well even if the source and target representation can be largely different.**
>
> **Response D:**
>
> - This is an excellent suggestion. Due to resource limitations, we were unable to conduct these experiments at the time of submission. We will try our best to add these experiments to the revised version.
>
> **Question E: How do the tokens retrieved from the** **auxiliary datastore and original datastore overlap? I'm curious how unique tokens can be retrieved by using the auxiliary datastore.**
>
> **Response E:**
>
> - Note that the auxiliary datastore is constructed from the top $m$ similar sentence pairs **ranked by the source-side similarity** between the input sentence and the source side of the sentence pairs. Therefore, we will recall different entries from these two datastores even if the same query is used. Experiments on the development set of the IT dataset reveal that only 7.38% of tokens retrieved from the auxiliary datastore overlap with those from the original datastore. Here, "overlap" means the two tokens are identical and from the same sentence pairs. This result indicates that the auxiliary datastore has the potential to (1) introduce more unique tokens and (2) recall complementary reference for the same tokens. In conjunction with our ablation study results on this module, this underscores the importance of considering source context when constructing datastore.
>
> **Question F: I understand that your method constructs the auxiliary datastore for each sentence. How heavy is it to construct?**
>
> **Response F:**
>
> - In our implementation, for an input source sentence, we retrieve $m$ similar sentences to construct the auxiliary datastore (to ensure enough entries for retrieval, we usually set $m$=$k$, where $k$ is the number of entries retrieved). As shown in Section 5.1 of our paper, its impact on efficiency is negligible. This is because the size of the auxiliary datastore is about 4-5 orders of magnitude smaller than different from the original datastore, for instance 285 vs 3,602,862 on the IT dataset.
>
> **Question G: It might be nice to cite this concurrent work, which uses source-side information to filter the datastore. https://aclanthology.org/2023.acl-long.10/**
>
> **Response G:**
>
> - We will add an introduction to this related work in the revised version. Thanks for the suggestion.
>
> &nbsp;
>
> [1] Zheng X, Zhang Z, Guo J, et al. Adaptive Nearest Neighbor Machine Translation[C]//Proceedings of the 59th Annual Meeting of the Association for Computational Linguistics and the 11th International Joint Conference on Natural Language Processing (Volume 2: Short Papers). 2021: 368-374.
>
> [2] Craswell, N. (2009). Precision at n. In: LIU, L., ÖZSU, M.T. (eds) Encyclopedia of Database Systems. Springer, Boston, MA.
>
> [3] Craswell, N., Robertson, S. (2009). Average Precision at n. In: LIU, L., ÖZSU, M.T. (eds) Encyclopedia of Database Systems. Springer, Boston, MA.
>
> [4] Beitzel, S.M., Jensen, E.C., Frieder, O. (2009). MAP. In: LIU, L., ÖZSU, M.T. (eds) Encyclopedia of Database Systems. Springer, Boston, MA.

---

### Official Review · Reviewer_aAZW · 2023-08-04

**Soundness:** 3

**Excitement:**

2: Mediocre: This paper makes marginal contributions (vs non-contemporaneous work), so I would rather not see it in the conference.

**Missing References:**

Some related work in general context-aware NMT. There are papers by Sennrich, Voita, and many others.

**Paper Topic And Main Contributions:**

The paper presents a method for using source context in nearest neighbor machine translation (kNN-MT). The authors compare their proposed method against other kNN-MT models and the vanilla Transformer model on several languages and domains, and report constant improvements in BLEU scores.

**Reasons To Accept:**

Perhaps an interesting add-on to the series of previous kNN-MT papers.

**Reasons To Reject:**

Training models for high-resource languages like English, German and Spanish using tiny fractions of the available data and reporting any improvements over such models is not appropriate. All datasets used have few hundred thousands of parallel sentences while both EN-DE and EN-ES have close to a billion parallel sentences available on Opus. This makes the reader question if the proposed method would have changed anything at all when trained on reasonable scale data. There's even no need to go large since strong models can be trained in days on a single GPU with at least a few tens of millions parallel sentences.

It is very unclear what are the `four commonly benchmark domain datasets` mentioned on line 257. They are not common at all and the citation from line 255 is not the correct source here. The authors also do not specify translation directions or even languages for the four datasets. Only for JRC they do.
Would this be the correct citation https://aclanthology.org/W17-3204/ ? In that case, the *law* part there is the same JRC Acquis.

In addition, when reporting BLEU score improvements of around 1-2 it would always be a good idea to perform statistical significance testing to make sure that the results are not just by chance. This has not been done in this paper. Not that it would offset using tiny amounts of training data for resource-rich languages, but would still make the results more credible.

**Reproducibility:**

3: Could reproduce the results with some difficulty. The settings of parameters are underspecified or subjectively determined; the training/evaluation data are not widely available.

**Reviewer Confidence:**

4: Quite sure. I tried to check the important points carefully. It's unlikely, though conceivable, that I missed something that should affect my ratings.

**Typos Grammar Style And Presentation Improvements:**

Having the BLEU score improvement mentioned in the abstract does not say much without knowing the involved languages, translation direction and domain. Either add more details or remove. The introduction is also not the best place for mentioning it (line 103).

---

> ### Author Rebuttal · Authors · 2023-08-29
>
> **Question A: Experiments on larger datasets.**
>
> **Response A:**
>
> - Following the settings in recent strong baselines[1, 2], we conduct experiments on the four datasets, i.e., IT, Medical, Koran, and Law, to make our work directly comparable with these recent strong baselines. We fully agree with the reviewer's opinion that experiments on larger datasets will make the results more solid. Therefore, we conduct experiments on the Subtitles dataset[3], and use the split version from Aharoni and Goldberg[4], whose training set consists of **14M** parallel sentence pairs and is leveraged in the original kNN-MT paper[5].  **In summary, the results show that our method introduces significant improvements under both the Domain Adaptation and In Domain settings on this larger dataset**, justifying the effectiveness of our work. Note that we try our best but fail to provide experimental results under the WMT19 data setting[5] due to both resource limitation and reproduction problems, which will be explained at the end of this response.
>
> - **Dataset**: We conduct the experiments on the Subtitle dataset[3], which consists of 14M parallel sentence pairs, two orders of magnitude larger than the four datasets we originally used. The translation direction is de=>en.
>
> - **Results under Domain Adaptation Setting:** Experimental results on two versions of base models will be reported:
>
>   - The WMT19 de=>en direction winner model[6] (**WMT19-winner**), which is the standard setting leveraged in kNN-MT baselines;
>
>   - A model finetuned from WMT19-winner on the Subtitle training set (**WMT19-finetuned**), which simulates the scenario where the training data of the target domain is large.
>
>   The following table shows the results on the Subtitle evaluation set. We can see that the two strong kNN-MT baselines are still effective on the larger dataset. And our method demonstrates statistically significant improvement  ($p < 0.05$, indicated by "*") over the two baselines.
>
>   | Model                  | BLEU           |
>   | ---------------------- | -------------- |
>   | WMT19-winner           | 29.58          |
>   | Adaptive kNN-MT        | 31.23          |
>   | Adaptive kNN-MT + Ours | 32.08* (+0.85) |
>   | Robust kNN-MT          | 31.68          |
>   | Robust kNN-MT + Ours   | 32.39* (+0.71) |
>
>   Moreover, from the following table we can observe that finetuning a base model on the large training set of the target domain can achieve better results (29.58 v.s. 31.75). Both kNN-MT baselines surpass the performance of the stronger base model, which underscores the efficacy of the kNN-MT approaches. Furthermore, our methods also outperforms these kNN-MT baselines significantly ($p < 0.05$).
>
>   | Model                  | BLEU           |
>   | ---------------------- | -------------- |
>   | WMT19-finetuned        | 31.75          |
>   | Adaptive kNN-MT        | 32.66          |
>   | Adaptive kNN-MT + Ours | 33.35* (+0.69) |
>   | Robust kNN-MT          | 32.95          |
>   | Robust kNN-MT + Ours   | 33.57* (+0.62) |
>
>   **Above all, the results on larger datasets justifies the effectiveness of our method.**
>
> - **Results under In Domain Setting:** As the WMT19-finetuned model is finetuned on the Subtitle training set, it can also be viewed as an in-domain model of Subtitle. Consequently, the results in the second table in last section also justify that our method is effective for the in domain setting with large training data. Additionally, we trained a Transformer model from scratch using the Subtitle training set. The findings from this experiment, presented in the subsequent table, align with our earlier observations. Therefore, we think we can safely conclude that our method is also effective for the in domain setting.
>
>   | Model                  | BLEU           |
>   | ---------------------- | -------------- |
>   | Transformer            | 29.96          |
>   | Adaptive kNN-MT        | 31.38          |
>   | Adaptive kNN-MT + Ours | 32.47* (+1.09) |
>   | Robust kNN-MT          | 31.91          |
>   | Robust kNN-MT + Ours   | 32.66* (+0.75) |
>
> - **Comments on WMT19 experiments**: The WMT19 dataset leveraged in the orginal kNN-MT paper[5] is a much larger dataset. However, we fail to reproduce the result due to both resource and code problems. On the one hand, the raw datastore is about 2TB, and the corresponding index built by [faiss](https://github.com/facebookresearch/faiss) for retrieval would exceed 50GB, making it impossible to be loaded into our available GPUs. On the other hand, the code provided by the authors of kNN-MT[5] does not contain the code to reproduce the results on WMT19. We find other researchers also faced the above problems and struggle to reproduce the results (https://openreview.net/forum?id=uu1GBD9SlLe, ICLR 2023).
>
> **Question B:** **More details on the four domain datasets.**
>
> **Response B:**
>
> - These datasets are from [3]. We follow the same settings as Adaptive KNN-MT[1] and Robust kNN-MT[2], and use the split version from Aharoni and Goldberg[4]. The translation directions are German to English.
>
> **Question C: Overlap between the Law and JRC-Acquis datasets.**
>
> **Response C:**
>
> - Thank you for pointing out the overlap between the Law dataset and the JRC Acquis dataset. We will annotate this in our revised version, but we believe this may not affect our argumentation. In the Domain Adaptation setting, we use the WMT19 winner model as our base model, build a datastore using the Law dataset, and compare the performance differences between several baselines and the proposed method. This is to demonstrate that source context can enhance the Domain Adaptation capability of the kNN-MT model. In the In-Domain setting, we use the Transformer model trained on the JRC Acquis dataset as our base model and continue to construct a datastore using the JRC Acquis dataset. This is to justify that context can enhance the In-Domain capability of the kNN-MT model. It can be seen that due to different settings, the overlap between the Law and JRC Acquis datasets does not affect our argumentation.
>
> **Question D: Lack of significance test for experimental results.**
>
> **Response D:**
>
> - Thank you for the reminder. We have added a significance test as shown on the following table. The symbol * represents significant improvement (p<0.05). For all comparisons except for the Koran dataset, our proposed method is significantly better than the strong baselines, demonstrating the effectiveness of our method. We explained the reasons for insignificant performance improvement on the Koran dataset in Section 4.2 of our original submission.
>
>   | Model                  | IT     | Medical | Koran | Law    |
>   | ---------------------- | ------ | ------- | ----- | ------ |
>   | Adaptive kNN-MT        | 47.74  | 56.12   | 20.31 | 62.87  |
>   | Adaptive kNN-MT + Ours | 49.31* | 57.72*  | 20.38 | 64.14* |
>   | Robust kNN-MT          | 48.78  | 57.11   | 20.50 | 63.61  |
>   | Robust kNN-MT + Ours   | 49.66* | 57.78*  | 20.72 | 64.52* |
>
>   | Model                  | Es->En | En->Es | De->En | En->De |
>   | ---------------------- | ------ | ------ | ------ | ------ |
>   | Adaptive kNN-MT        | 66.01  | 64.09  | 64.59  | 58.54  |
>   | Adaptive kNN-MT + Ours | 67.87* | 65.53* | 65.67* | 60.31* |
>   | Robust kNN-MT          | 67.08  | 64.97  | 64.93  | 59.52  |
>   | Robust kNN-MT + Ours   | 67.95* | 65.60* | 65.73* | 60.48* |
>
> **Question E: Lack of relevant references like context-aware NMT.**
>
> **Response E:**
>
> - Thank you for the reminder. We will add relevant context-aware references to the revised version.
>
> &nbsp;
>
> [1] Zheng X, Zhang Z, Guo J, et al. Adaptive Nearest Neighbor Machine Translation[C]//Proceedings of the 59th Annual Meeting of the Association for Computational Linguistics and the 11th International Joint Conference on Natural Language Processing (Volume 2: Short Papers). 2021: 368-374.
>
> [2] Jiang H, Lu Z, Meng F, et al. Towards Robust k-Nearest-Neighbor Machine Translation[C]//Proceedings of the 2022 Conference on Empirical Methods in Natural Language Processing. 2022: 5468-5477.
>
> [3] Koehn P, Knowles R. Six Challenges for Neural Machine Translation[C]//Proceedings of the First Workshop on Neural Machine Translation. 2017: 28-39.
>
> [4] Aharoni R, Goldberg Y. Unsupervised Domain Clusters in Pretrained Language Models[C]//Proceedings of the 58th Annual Meeting of the Association for Computational Linguistics. 2020: 7747-7763.
>
> [5] Khandelwal U, Fan A, Jurafsky D, et al. Nearest Neighbor Machine Translation[C]//International Conference on Learning Representations. 2021.
>
> [6] Ng N, Yee K, Baevski A, et al. Facebook FAIR’s WMT19 News Translation Task Submission[C]//Proceedings of the Fourth Conference on Machine Translation (Volume 2: Shared Task Papers, Day 1). 2019: 314-319.

---

### Meta-Review · Area_Chair_EGR2 · 2023-09-21

**Recommendation:** 3

**Metareview:**

The work augments kNN-MT with source context enhancement, including a distance calibration module. Such a method has the benefit of being easily integrated into existing kN-MT setups, and the authors test this across various settings. The reviewers express some concerns, such as more competitive baselines (especially in non-artificial low-resource settings), and the authors present interesting results in the rebuttal. I'd recommend the authors also include some sense of the variation in BLEU, to better understand if improvements like +0.6 are significant.

---

### Decision · Program_Chairs · 2023-10-07

**Decision:**

Accept-Main

**Comment:**

The work augments kNN-MT with source context enhancement, including a distance calibration module. Such a method has the benefit of being easily integrated into existing kN-MT setups, and the authors test this across various settings. The reviewers express some concerns, such as more competitive baselines (especially in non-artificial low-resource settings), and the authors present interesting results in the rebuttal. I'd recommend the authors also include some sense of the variation in BLEU, to better understand if improvements like +0.6 are significant.